# A Repetitive *Acipenser gueldenstaedtii* Genomic Region Aligning with the *Acipenser baerii* IGLV Gene Cluster Suggests a Role as a Transcription Termination Element Across Several Sturgeon Species

**DOI:** 10.3390/ijms252312685

**Published:** 2024-11-26

**Authors:** Alexander V. Chouljenko, Brent A. Stanfield, Tetiana O. Melnyk, Ojasvi Dutta, Vladimir N. Chouljenko

**Affiliations:** 1Department of Food, Bioprocessing and Nutrition Sciences, North Carolina State University, Center for Marine Sciences and Technology, Morehead City, NC 28557, USA; avchoulj@ncsu.edu; 2Department of Pathobiological Sciences, Louisiana State University School of Veterinary Medicine, Baton Rouge, LA 70803, USA; bstanf5@lsu.edu (B.A.S.); tmelnyk@lsu.edu (T.O.M.); odutta@lsu.edu (O.D.); 3Division of Biotechnology and Molecular Medicine, Louisiana State University School of Veterinary Medicine, Baton Rouge, LA 70803, USA

**Keywords:** sturgeon, caviar, DNA, repetitive sequences, PCR, transcription termination

## Abstract

This study focuses on the common presence of repetitive sequences within the sturgeon genome that may contribute to enhanced immune responses against infectious diseases. A repetitive 675 bp VAC-2M sequence in Russian sturgeon DNA that aligns with the Siberian sturgeon IGLV gene cluster was identified. A specific 218 bp long portion of the sequence was found to be identical between *Acipenser gueldenstaedtii*, *A. baerii* and *A. stellatus* species, and NCBI blast analysis confirmed the presence of this DNA segment in the *A. ruthenus* genome. Multiple mutated copies of the same genomic region were detected by PCR analysis, indicating that different versions of this highly repetitive sequence exist simultaneously within the same organism. The selection toward specific genetic differences appears to be highly conserved based on the sequence variations within DNA originating from fish grown at distant geographical regions and individual caviar grains from the same fish. The corresponding *A. baerii* genomic region encompassing the 357 bp DNA sequence was cloned either ahead or after the human cytomegalovirus immediate early promoter (HCMV-IE) into a pBV-Luc reporter vector expressing the luciferase gene. The DNA segment significantly reduced luciferase expression in transient transfection/expression experiments. The results indicate that this genomic region functions as a transcription termination element that may affect antibody production in sturgeons.

## 1. Introduction

Sturgeons belong to the order Acipenseriformes and include 25 caviar-producing species, of which 17 are members of the *Acipenser* genus [1]. All the existing sturgeons are polyploid and have an extremely complicated structure of the genome which underwent a series of genome polyploidizations from the original 2n~60 chromosomes precursor to derive tetraploid (4n~120 chromosomes), octoploid (8n~240 chromosomes) and sixteen-ploid (16n~480 chromosomes), that are corresponding to three rounds of whole genome duplications (WGDs) discovered for these species [2,3]. The viable juvenile sturgeon hybrids with the highest known for vertebrate’s chromosome count (477–520) were also reported [4]. Siberian sturgeon (*A. baerii*) and Russian sturgeon (*A. gueldenstaedtii*) are the species most grown in aquaculture and both have 8n ploidy [5] while the smaller sterlet sturgeon (*A. ruthenus*) that is known to hybridize naturally with other sturgeon species [6] has 4n ploidy.

Until the recent advances of the Next-Generation Sequencing (NGS) technology, the sturgeon genome complexity was an obvious impediment, but currently, the very well-annotated whole genome sequence of the *A. ruthenus* (GenBank taxid 7906) together with the genomic data sets for *A. baerii* (GenBank taxid 27689) and *A. gueldenstaedtii* (GenBank taxid 7902) are also available. The rapid progress of the whole genome sequencing technology has added an extra incentive to the discovery and analysis of the repetitive DNA sequences that are widely present in all eukaryotes. They play critical roles in driving evolution by inducing genetic variation, regulating gene expression and can account for 25–50% of the eukaryotic genome [7]. An estimated 40.3% of the *A. ruthenus* genome was found to be represented by the repeats [8]. The first study directly revealing the exact genomic location of genes encoding the variable light (VL) segments of the immunoglobulin light chain (IGLV) of the *A. baerii* was reported by Lundqvist and Pilström [9]. The relevant GenBank entry describes a partial sequence of the immunoglobulin light-chain variable region of the *Acipenser baerii* (AJ245365). Interestingly, within the ~8 kb of sequence preceding the *A. baerii* IGLV gene, there is a recognizable primer binding site (PBS) for tRNA-Leu, a GAG-encoding region, then a protease-encoding region (PRO), followed by an IN-encoding region strongly suggesting about the possible LTR retrotransposon location [10]. Sequence-specific DNA-binding transcriptional regulators comprise one critical type of control for determining whether and to what extent each gene in the genome is transcribed. The large portion of repetitive *A. gueldenstaedtii* -derived VAC-2M DNA fragments (434 out of 675 nts) aligned with the respective sequence of *A. baerii* with 89% identity without any gaps and is located within the ~6 kb DNA fragment preceding the IGLV gene. The region was specific for at least two more sturgeon species (*A. ruthenus* and *A. stellatus*) and contained a cluster of three sequence motifs (GATTAT) that can potentially serve as a DPRX binding site [11]. DPRX is a member of the paired (PRD)-like homeobox gene family of transcription factors that is expressed in early embryos and is thought to play a role in the regulation of embryo genome activation and preimplantation embryo development [12,13].

The primary objective of the study was to investigate the potential regulatory role of the highly repetitive sequence VAC-2M that is found in several *Acipenser* species and located within a genetic locus preceding an immunoglobulin light chain variable region gene cluster.

## 2. Results

### 2.1. Repetitive Elements Within the Acipenser gueldenstaedtii Genome

Two repetitive DNA fragments (VAC-1M and VAC-2M) within the *A. gueldenstaedtii* genome were identified by random PCR amplification using a short 10 bases long oligonucleotides and DNA from five adult females (F) and five adult males (M) fish grown at the Marshallberg Farm (MF), North Carolina, during our attempts to delineate sex-specific genetic markers for the Russian sturgeons. Some more details regarding this part of the study are shown in the Appendix A. Briefly, DNA from the four out of five adult males used (M2 through M5, Figure 1) contained a highly repetitive 735 bp long DNA sequence (715 bp without counting 10 + 10 bases belonging to the random oligo sequence used for amplification). For two out of five adult females (F2 and F4, Figure 1), the respective region contained 39 bp deletion (the PCR product was 696 bp long), and three out of five females (F1, F3, and F5, Figure 1) did not produce PCR products of comparable size, which was most likely due to a larger deletion or because this genomic region was completely missing. The DNA fragment was designated as VAC-1M, and specific primers were designed based on this sequence after cloning and sequencing (Appendix A). At least for the adult fish, we could clearly differentiate female-derived from the male-derived DNA using these primer sets (Figure 2). However, one adult male (M1) contained both 735 bp and 696 bp versions of this genomic area (Figure 1 and Figure 2), providing us with the initial experimental data suggesting the simultaneous existence of different mutated and unchanged DNA copies of the specific genomic region within the same fish. In the end, we did not receive the results we hoped for regarding the highly repetitive VAC-1M sequence use as a genetic marker for sturgeon sex determination (see Appendix A). However, the data collected provided an opportunity to explore a completely new research direction toward the goal of better understanding the role of the repetitive elements within the sturgeon genome in general.

The conventional sequencing of 16 male-derived cloned DNA fragments (four clones each representing DNA samples M2 through M5) was performed. The results revealed that 14 out of 16 sequences designated as VAC-1M were similar and exhibited only a single nt substitution between the samples. The consensus sequence was generated, and NCBI blast search results against the entire GenBank database revealed the multiple high scores matches within different chromosomes of the best annotated whole genome sequencing data set of the *A. ruthenus*, thus further confirming the repetitive nature of this DNA fragment. Interestingly, two other clones (designated as VAC-2M) that derived from the adult M5 DNA sample were also almost identical repetitive sequences but completely different from the VAC-1M. They were 695 bp long (675 bp without 20 nts provided by the random oligo sequence contribution) and, most likely, co-migrated on the gel together with VAC-1M before being used for cloning.

The most fascinating match during the similar NCBI search revealed that a large portion of this DNA fragment aligned significantly to the *Acipenser baerii* genomic region reported almost 25 years ago (GenBank accession number AJ245365) and contained the partial IGLV gene for immunoglobulin light chain variable region, exons 1–2 [9]. The first 434 bp out of 675 bp long VAC-2M sequence aligned with 89% identity (388/434, with 0 gaps) against nts 3130–3563 of the reported *A. baerii* IGLV gene cluster (“old” *A. baerii*, total 10,334 bp DNA fragment with a presumable initiation codon for exon 1 located at nt position 8806). NCBI BLAST search results against the best-annotated whole genome sequencing data set of *A. ruthenus* revealed the presence of this DNA fragment across multiple chromosomes. Some copies exhibited up to 97% sequence identity with only nucleotide substitutions and no gaps due to deletions or insertions. The respective available *A. gueldenstaedtii* sequencing data set does not provide individual chromosome distribution; however, multiple copies of the targeted DNA fragment were also scattered within multiple assembled DNA contigs.

Primers located within a matching 434 bp portion of the cloned sequence (M5-1F/M5-1Rn, Appendix A) were designed to determine whether PCR amplification of this repetitive element was specific for the Russian sturgeon in general and not limited to the random adult male fish grown at the Marshallberg Farm. DNA templates were isolated from the individual caviar grains that originated from geographically distant sturgeon fish farms (a farm in Israel (IF) vs. MF, Figure 3). Most of the PCR products looked like smears after separation on the 1.5% agarose gel with the highest yield area located around the 400 bp region as would be expected based on the primer’s location. Surprisingly, a distinct DNA fragment that migrated just below the 200 bp band of the ladder was observed only for the IF-derived DNA samples (Figure 3, lanes 1–9). The corresponding bands from the first four lanes were cut from the gel, purified and used for cloning into pcDNA 3.1/V5-His-TOPO vector. A total of eight clones (two clones represented each individual caviar graine) were sequenced using vector-specific primers (CMV-for, Appendix A). All cloned DNA fragments were almost identical and similar in size (192 bp long) and were designated as Bottom (B) fragments. Only different single nt substitutions were detected within three out of eight clones sequenced, and a consensus sequence for the B fragment was generated and is shown side by side with the respective *A. baerii* region (Figure 4A,B). A set of the two new internal primers (B-IF-for/B-IF-rev, Figure 4, Appendix A) were designed based on the sequence of a 192 bp long PCR product to validate a specific amplification of the smaller size PCR products (139 bp vs. 168 bp long, as would be expected from the respective *A. baerii* sequence). DNA was extracted (three individual caviar grains from each of the five female sturgeons provided by Marshallberg Farm were used), yielding a total of 15 DNA samples from the MF caviar. Additionally, the same DNA from the first four of IF-derived caviar samples as shown in Figure 3 were selected for amplification using B-IF-for/B-IF-rev primers. Surprisingly, all MF-derived PCR products (Figure 5, lanes 5–19) generated a similar and what looked like 168 bp long band on the gel (consensus sequence for the 168 bp products is shown in Figure 4C) in contrast to the 139 bp long PCR products that originated from the IF-derived fish grown in Israel (Figure 5, lanes 1–4). Bands of interest were cut from the gel, purified, and both strands of the PCR products were sequenced directly using amplification primers. All IF-derived PCR products as well as MF-derived samples from females I, II, and V generated very good quality sequences that were 139 bp and 168 bp long, respectively. PCR products from the MF-derived females III and IV failed an initial direct sequencing attempt and were sequenced after cloning using a vector-based primer (CMV-for, Appendix A). We observed the presence of both the 168 bp and 139 bp variants in DNA extracted from different individual caviar grains originating from the same fish, suggesting that selection toward either mutation may commence during the earliest stages of development.

### 2.2. Specific Mutations Within the Corresponding Repetitive Genomic Regions of the Russian and Siberian Sturgeon Species

The sampler set that included Sevruga, Siberian, and Russian sturgeon caviar has been acquired commercially from Evans Fish Farm (EF), FL, USA. DNA was isolated from the total of 12 each individual grains of the EF-derived Siberian and Russian sturgeon caviar and subjected to PCR using B-IF-for/B-IF-rev primers. All 12 EF-derived Russian sturgeon DNA samples generated a distinct 139 bp long band on the gel (only two representatives are shown in Figure 6, lanes 11–12). The EF-derived consensus 139 bp sequence was identical to the one produced by the IF-derived DNA samples (only two representatives are shown in Figure 6, lanes 7–8). In contrast, MF-derived Russian sturgeon DNA samples produced mostly 168 bp long PCR products (Figure 6, lanes 9–10). The respective *A. baerii* DNA samples are shown in Figure 6 lanes 1–6 and 13–18. Multiple individual clones were sequenced, and the results confirmed that almost all Siberian sturgeon-specific DNA samples generated both 139 bp and 168 bp products. DNA sequences originated from the Russian sturgeon fish of very different geographical origin (Israel vs. NC, USA and vs. FL, USA) have been compared with the respective Siberian sturgeon-derived 139 bp and 168 bp long products from the recent EF-derived DNA and the corresponding 168 bp long Siberian sturgeon genomic region reported 25 years ago (Figure 7).

Overall, we have detected a 139 bp version of this genomic region for all *A. gueldenstaedtii* samples grown at three different fish farms (IF, MF, and EF) as well as for *A. baerii* originated at the EF. They all have almost identical sequences except a single nucleotide substitution specific to *A. baerii* (Figure 7, nt position 140), and all have the same 29 bp deletion if compared to the original 168 bp long sequence of this region. The 168 bp version was detected for Siberian sturgeon from the EF and only Russian sturgeon fish grown at the MF. The sequences for both species are identical if compared to each other, and they all have the same mutations at nt positions 25 and 126 in comparison to the 139 bp variation (Figure 7). However, both 168 bp and 139 bp long sequences are very different from the original *A. baerii* sequence for this region reported 25 years ago. They have five more identical nt substitutions located outside of the primers used for amplification (nts 36, 39, 45, 51, and 138 as shown in Figure 7). There are two more mutations that were introduced by forward amplification primer (nts 13 and 16) and 13 mutations introduced by the reverse primer (shown in bold as small, underlined letters in primer sequence). Importantly, both oligos were originally designed based on the cloned and specific sequence of the B fragment (see Figure 3 and Figure 4). Multiple DNA samples from skin swabs originated from fish of different ages grown at the MF were tested using B-IF-for/B-IF-rev primers to further validate the 168 bp/139 bp types of variations. The results were very similar to the MF caviar-derived DNA testing. Most of the samples generated 168 bp PCR products while some, as shown in Appendix A, contained both versions, indicating that the process of mutation selection is independent from the fish ages, most likely not random, and is ongoing.

### 2.3. Mutated and Unchanged Copies of the Repetitive Genomic Region Exist Simultaneously Within the Same Fish Regardless of the Sturgeon Species

B-IF-for/B-IF-rev primer pairs were specifically designed based on the sequence from the cloned B fragment. The forward primer contained only two nt substitutions if compared to the corresponding *A. baerii* sequence (Figure 4B). Importantly, the last five bases located at the 3′ end of the primer that are the most critical for the template binding were identical. The reverse primer was much more different and contained a total of 13 nts out of 19 that did not match the respective “old” *A. baerii* sequence, but the last 7 nts out of 9 were also identical, thus providing an opportunity to generate PCR products. A different reverse primer located downstream (A ba 3411R, Figure 4B, Appendix A) was designed to produce a 218 bp long PCR product if used in combination with the B-IF-for primer. The same Russian and Siberian sturgeon caviar DNA samples that produced 168 bp/139 bp PCR products were combined with three additional DNA samples isolated from the available *A. stellatus* individual caviar grains. All DNA samples generated PCR products of the expected 218 bp long size regardless of the sturgeon species (Figure 8). The direct sequencing of both strands of these DNA fragments revealed that all analyzed sequences were identical and absolutely the same as the respective *A. baerii* sequence published 25 years ago. NCBI blast search results against the best annotated whole genome sequencing data set of *A. ruthenus* revealed the presence of this DNA fragment within the multiple chromosomes with a total number of copies estimated to be at least 800 and with approximately 3% of them having 100% sequence identity. The respective available *A. gueldenstaedtii* sequencing data set does not provide individual chromosome distribution, but multiple copies of the targeted DNA fragment were also scattered within the assembled DNA contigs.

### 2.4. Both Variable Domain and Immediately Adjacent Genomic Region of the Repetitive Sequence Are Important Regulatory Elements That Significantly Decreased Expression of the Luciferase Reporter Gene

MEME suite (Motif-based sequence analysis tools) allows identification of the specific motifs located within any given sequence [14]. The scan of the entire *A. baerii* IGLV gene cluster (10,334 bp) revealed the presence of a total of eight sequence motifs (GATTAT) that can potentially serve as a DNA binding site for the transcription termination [11]. The first two motifs were located at the respective nt positions 1222–1227 and 1315–1320, while the next three were clustered (nts 3370–3375, 3464–3469, and 3541–3546) within the short stretch of the sequence that was located completely within the boundaries of the 357 bp long cloned portion of the repetitive region (*A. baerii* nts 3202–3558 or ABter domain). This domain included a variable for different sturgeon species 160 bp segment (*A. baerii* nts 3202–3361) that constituted almost all sequences of the 168 bp long PCR products and an immediately adjacent region containing three motif sequences mentioned above (*A. baerii* nts 3361–3558 or **∆**ABter domain). The corresponding portion of the VAC-2M repeat sequence contained a total of 20 nt substitutions in comparison to the respective “old” *A. baerii* region. Remarkably, none were detected inside of any of the three potential DNA binding sites. Another two more motif sequences were located just outside from the cloned repeat region (nts 3883–3888 and nts 4723–4778), and the last one was detected at the *A. baerii* nt positions 10,043–10,048. Overall, five out of eight potential DNA binding sites were found to reside within the ~5.6 kb sequence preceding the IGLV gene cluster.

It is worth noting that multiple clones that were generated during an assembly of ~5.6 kb of the *A. baerii* genomic region exhibited a very high degree of variability with mostly nt substitutions or single base insertions/deletions in comparison to the “old” *A. baerii* sequence. Only 1 out of the 12 clones analyzed (clone 9) had a restriction profile similar to the published *A. baerii* sequence [9]. This clone also contained mutations that changed the presumable IGLV tata-box (contained TTTAA instead of TATAA) as well as mutation/insertion within the first adjacent putative E-box (contained CACTTTGCAT instead of CACTTGCAT). At the end, clone 9 was selected to be the origin for some of the constructs used for luciferase expression experiments. Both the DNA fragment covering the putative “old” *A. baerii* promoter sequence without any new mutations (569 bp DNA fragment was ordered separately as a G-Block) and the respective sequence from clone 9 (nts 8337–8806) were cloned into pBV-Luc plasmid (constructs 7 and 8, respectively, see the Materials and Methods section).

The original pBV-Luc plasmid (construct 1) was specifically designed to contain a pause site before a multiple cloning site linker followed by a luciferase reporter gene with an original aim to decrease the luciferase expression background. A total of three independent transfections were performed, and one of the side-by-side comparison results (plated in quadruplicates for the luciferase assay) is shown in Figure 9A,B. The transfection results using constructs 1 through 8 (see the Materials and Methods section for details) revealed that the pBV-Luc control plasmid generated the highest overall level of the luciferase expression (an average of 1306 U vs. 490 U produced by different background controls; see Figure 9A). All three plasmids containing the entire ~5.6 kb sequence preceding IGLV exon 1 (constructs 2 through 4) generated no luciferase expression with numbers consistently even lower than ones for the background controls. The deletion of the 160 bp portion of the repeat (*A. baerii* nts 3202–3361 removed, construct 5) resulted in a very similar outcome. The complete repeat deletion (*A. baerii* nts 3202–3558 removed, construct 6) generated the luciferase signal that was comparable to the level produced by the plasmids containing different versions of the cloned native IGLV promoter (constructs 7 and 8) but was still almost two times below the respective level for the control pBV-Luc plasmid.

To study what specific role the repetitive portion of the targeted region plays in the observed phenomenon, we have constructed an additional positive control plasmid for luciferase expression by having a DNA fragment-encoding HCMV-IE promoter cloned into pBV-Luc (construct 9). In the next step, the ABter domain was cloned into both pBV-Luc and pBV-Luc-CMV after the pause site but just before the promoter sequence in the (+) orientation (constructs 10 and 11, respectively). Two more plasmids contained the same domain cloned after CMV but before the luciferase gene in both orientations (− and +, constructs 12 and 13, respectively). Another set of four plasmids was similarly constructed but contained the **∆**ABter domain without the variable portion of the repeat (see Materials and Methods).

There are a total of 18 different plasmids (construct #18 has EGFP under control of the HCMV-IE promoter to monitor the transfection efficiency) that were used for transfection. ABter cloning into pBV-Luc generated luciferase expression numbers almost two times higher than pBV-Luc alone (average of 2029 U vs. 1306 U). The cloned HCMV-IE promoter sequence increased luciferase expression more than 200 times in comparison to the pBV-Luc plasmid without the promoter. The ABter domain cloned before HCMV-IE resulted in an approximately 1.6-fold reduction in luciferase expression, while the same sequence cloned after HCMV-IE reduced expression much more efficiently. Importantly, both orientations of the cloned ABter sequence exhibited a decrease in the luciferase expression by approximately 19 and 14-fold for the (−) and (+) strands, respectively. Similar cloning of the **∆**ABter domain before HCMV-IE decreased luciferase expression by approximately 1.4 times. The same sequence cloned after HCMV-IE reduced luciferase expression only by 5.0 and 4.7 times for the (−) and (+) strands, respectively (Figure 9B).

## 3. Discussion

The current study is focused on the role of the highly repetitive sequence VAC-2M that was originally detected using Russian sturgeon DNA. The sequence analysis of this genomic region specific for Russian and Siberian sturgeon species revealed conservation as well as significant genetic differences including multiple single nucleotide changes and a large deletion in comparison to the prototypic “old” *Acipenser baerii* sequence [11]. This genetic region was predicted to contain a cluster of three sequence motifs (GATTAT) that can potentially serve as DPRX binding sites that are associated with the regulation of embryo genome activation and preimplantation embryo development [12,13]. Both repetitive VAC-1M and VAC-2M sequences were originated from the male adult Russian sturgeons grown at the Marshallberg Farm, North Carolina, using random amplification PCR. Specific primers were designed based on the cloned VAC-1M DNA sequence and, at least for the adult fish, the use of these oligos allowed DNA-based sex differentiation. They were not viable, however, for the DNA-based sex diagnostic test when applied to the younger Russian sturgeon fish (Appendix A). Instead, the recently published AllWSex2 primers set [5] combined with our modification to include species-specific Ag49 primers [15] were validated to be used as a part of the multiplex PCR Russian sturgeon DNA-based sex diagnostic test (Appendix A).

The specific role of the VAC-1M repetitive element within the Russian sturgeon requires an additional investigation; however, the segment of the repeat covering the first 200–250 nts aligned with up to 96% identity with corresponding sequences within the 3′ regions of multiple genes encoded by different sterlet chromosomes. The highest scoring matches were detected for the *A. ruthenus* asc-type amino acid transporter 1-like mRNA (regulates amino acid transport within the cell); *A. ruthenus* copine-8-like transcript mRNA (may play a role in calcium-mediated intracellular processes); and *A. ruthenus* ubiquitin domain-containing protein 1-like mRNA (related to the ubiquitin–proteasome pathway). Similar NCBI search results using the first 150 bases of the VAC-2M sequence also revealed a high scoring match with up to 95% identity mostly within the 3′ regions of multiple genes specific to different *A. ruthenus* chromosomes. Interestingly, the first 434 bases of this repetitive sequence aligned with 89% identity (388/434, with 0 gaps) against *A. ruthenus* calcium/calmodulin-dependent protein kinase type 1D-like (LOC117419416), transcript variant X1 mRNA. In contrast to the VAC-1M repeat, the entire VAC-2M matching sequence was completely located inside of the respective ORF (gene location: nts1–5648; CDS location: nts 490–5088; VAC-2M matching: nts 2292–2725). Another high scoring alignment for the VAC-2M repeat was against the *A. baerii* partial IGLV gene for the immunoglobulin light chain variable region, exons 1–2 (89% identity (388/434, with 0 gaps) and was located within the ~5.6 kb sequence preceding the IGLV gene cluster). Specific primers within this 434 bp portion of the sequence were designed, and individual caviar grains from the fish grown at the geographically distant farms (Israel vs. North Carolina) were used for DNA isolation. The results confirmed the repetitive and specific for Russian sturgeon nature of the VAC-2M sequence; however, the distinct band on the gel whose size was smaller than expected was also observed only for the IF-derived caviar DNA templates (B fragment). After cloning and conventional DNA sequencing, it was determined that the B fragment corresponds to a portion of the VAC-2M sequence that was later found to represent the respective 139 bp type of sequence variation specific for this genomic region. A new set of internal primers was designed based on this cloned DNA fragment, and a 168 bp long PCR product was expected based on the respective “old” *A. baerii* sequence. Russian sturgeon DNA samples originated from the fish grown recently at three different fish farms (caviar samples from Israel, Evans Farm, Florida, and both caviar and skin swab samples from Marshallberg Farm, North Carolina, collected from fish of different ages, males and females) were used. Surprisingly, in addition to the predicted 168 bp long PCR product, we also detected the similarly highly mutated 139 bp long DNA fragment for all subjects tested. Commercially available caviar samples of the Siberian sturgeon grown at the EF were added to the mix, and both 168 bp and 139 bp variations were generated using DNA from the individual caviar grains. Overall, only 139 bp products were detected using the Russian sturgeon caviar DNA samples originated in Israel and at the EF. The Russian sturgeon caviar and skin swabs samples that originated at the MF as well as the EF Siberian sturgeon caviar DNA samples generated both 168 bp and 139 bp versions, strongly indicating that mutation selection is not random.

One of the issues associated with using commercially acquired caviar is an almost certain guaranteed outcome that different DNA samples isolated from the individual caviar grains will be derived from the same female fish. Marshallberg Farm provided caviar samples harvested from five different females, and similar 168 bp/139 bp variations were detected within different individual caviar grains that originated from the same fish. The sequence specific for the 168 bp type of variation was almost identical to the 139 bp version except they all contained the same 29 bp deletion. However, they were very different from the corresponding 168 bp “old” *A. baerii* sequence. A different reverse primer (A ba 3411R) located downstream was designed, and in combination with same forward (IF-B-for) primer, they were applied to the same DNA samples that generated 168 bp/139 bp variations to produce an expected 218 bp long PCR product based on the “old” *A. baerii* sequence. Some copies of this specific genomic region that included a 168 bp long DNA segment that was variable for different sturgeon species was found to be identical 25 years later for *A. baerii* and *A. stellatus* as well as for *A. gueldenstaedtii* grown recently at different and geographically distant fish farms. NCBI BLAST search results against the best annotated whole genome sequencing data set of *A. ruthenus* revealed the presence of this DNA fragment within the multiple chromosomes with total number of copies estimated to be at least 800 and with approximately 3% of them to have 100% sequence identity. The PCR outcome was very consistent regardless of the fish geographical origin, age, sex, source of DNA isolation (individual caviar grains vs. skin swabs) or sturgeon species (*A. gueldenstaedtii* vs. *A. baerii* vs. *A. stellatus*). The results effectively confirmed that different versions of this highly repetitive sequence exist simultaneously within the same organism, and the process of selection toward specific mutation is, most likely, not random. In addition, it is also ongoing based on the sequence variations within DNA derived from different individual caviar grains that originated from the same fish. Specific at least for the Russian and Siberian sturgeon species, an identical 29 bp deletion constitutes the main difference between the 168 bp vs. 139 bp types of variation. Otherwise, the sequences are very similar and represent a very mutated version of the “old” *A. baerii* region of the same size. Both the Israel farm and Evans Fish Farm, Florida, are historically more long-term established businesses, and the Russian sturgeon DNA samples that originated from these farms generated predominantly the 139 bp type of mutation. Marshallberg Farm is a relatively new location, and both 168 bp and 139 bp sequence variation types were detected, indicating that, most likely, the selection process goes from the “old” *A. baerii* through 168 bp and to the 139 bp type of sequence variations with the possibility of different copies coexisting within the same fish.

The pBV-Luc vector was specifically designed to contain the transcription termination element (pause site) located before the multiple cloning sites linker sequence followed by the reporter gene to reduce the background level of luciferase expression. Our initial transfection results using Vero cells and pBV-Luc-based plasmids (constructs 1 through 8) revealed that the highest level of luciferase reporter gene expression was detected for the pBV-Luc control plasmid. All constructs containing different versions of the entire ~5.6 kb sequence preceding the IGLV gene cluster (constructs 2 through 4) cloned after the natural pause site but before the luciferase gene generated no luciferase expression, and numbers were consistently below the background empty plate or irrelevant plasmid transfection controls. Vero cells are not considered to represent a homologous system for the *A. baerii* putative promoter testing, potentially explaining the low levels of luciferase expression generated by the respective plasmids 7 and 8. The results have demonstrated that the entire *A. baerii* ~5.6 kb DNA sequence preceding the IGLV gene starting from the repetitive region and up to the IGLV initiation codon is important to suppress luciferase expression, while the complete ABter domain removal (plasmid 6) has started the trend of expression level restoration.

To study the specific role of the ABter region in the observed phenomenon, we have constructed the positive control luciferase expression plasmid (construct 9) under regulation of the constitutive human cytomegalovirus-derived immediate early (HCMV-IE) promoter sequence. This promoter is one of the most utilized regulatory elements used for the vector’s construction and was validated to work efficiently in eukaryotic organisms, including fish [15,16,17]. The use of this plasmid for the transfection of Vero cells resulted in a more than 200-fold increase in luciferase expression in comparison to the original pBV-Luc plasmid. The ABter region is 357 bp long and represents the corresponding *A. baerii* genomic area homologous to the *A. gueldenstaedtii*-derived portion of the VAC-2M sequence. Some copies of the repeat containing the first 168 bp of this sequence were found to be variable at least for Russian and Siberian sturgeon species. Simultaneously, other copies representing the first 218 bp of the same region and specific for Russian, Siberian, Sevruga, and Sterlet sturgeon species were also found to be unchanged over 25 years. The next almost 200 bp long segment of the repeat (∆ABter) included three potential DNA binding motifs sequences (GATTAT). Importantly, this genomic area contained a total of 20 nt substitutions between the “old” *A. baerii* IGLV cluster sequence and the respective *A. gueldenstaedtii*-derived VAC-2M sequence, but none were located within the potential DNA binding sites. All mutations between the different sturgeon species were disproportionately clustered around these sites. Both (+) and (−) orientations of the ABter repeat cloned after the HCMV-IE promoter sequence resulted in a significant reduction in the reporter gene expression (by 14- and 19-fold, respectively). The ∆ABter portion of the repeat cloned similarly after HCMV-IE resulted in only 4.7- and 5.0-fold luciferase expression reductions for the (+) and (−) orientations, respectively. Similar ABter and ∆ABter cloning in the (+) orientation before the HCMV-IE sequence was much less efficient regarding the suppression of the luciferase expression level (1.6- and 1.4-fold reductions, respectively). The entire ABter domain, most likely, is required to serve as an effective transcription termination element.

Sturgeon genome complexity (4n-8n ploidy level) and the situation when different multiple copies of the repetitive genomic region do exist simultaneously within the same organism creates an enormous number of genetic variations to potentially allow a quick immune response to any of the outside triggers. It may also explain the extremely high mutation rate phenomenon observed during an assembly of the ~5.6 kb region preceding the IGLV gene even after exclusively using FailSafe polymerase for the overlap-extension PCR amplification. Different smaller DNA fragments used for an assembly of the final product could be derived from the different copies of the respective genomic region, thus creating multiple mutations within the final PCR product. Retrotransposons represent the most abundant form of repetitive DNA in the eucaryotic genomes, and a full-length LTR (Gmr1) was discovered within the sequence from the Atlantic cod, *Gadus morhua* [10]. Investigators have determined that the closest relative for this element is the respective Abr1 retrotransposon fragment from the Siberian sturgeon *A. baerii*. More importantly, within the ~8 kb sequence preceding the Siberian sturgeon IGLV gene, they found all the main features specific for the LTR location. This observation may indicate that this entire region of *A. baerii* is represented by the repetitive sequence, and the VAC-2M repeat first detected by us for the *A. gueldenstaedtii* is only part of the larger repeat DNA fragment. A recent review by Tower [18] discusses so-called selectively advantageous instability (SAI) as one or more components of a replicating system, such as the living cell. Short-lived transcription and signaling factors enable a rapid response to a changing environment, and turnover is critical for the replacement of damaged macromolecules. In summary, SAI promotes replicator genetic diversity and reproductive fitness by keeping both normal and mutated copies and may promote aging through the loss of resources and maintenance of deleterious alleles.

The study’s results reinforce the coexistence of different versions of the repetitive sequence in the same organism, indicating a non-random selection process driving specific mutations within DNA derived from distinct origins but from the same fish. Although the functional implications of these mutations remain unclear, the study highlights the potential role of these repetitive genomic regions in transcriptional regulation. Furthermore, our results indicate that this DNA segment can serve as a transcription termination element, warranting further exploration. Potentially, this regulatory element can be used as a biomarker if it can be associated with climate-resistant or infectious disease-resistant species. In addition, validation of this genetic element could find use in the production of transgenic fish with the desired characteristics.

## 4. Materials and Methods

### 4.1. Sturgeon DNA Samples Origin

*Acipenser baerii* and *Acipencer stellatus* caviar samples were acquired commercially from Evans Fish Farm (EF), Pierson, FL, USA. Two different caviar samples of *Acipenser gueldenstaedtii*, one originated from a fish farm in Israel (IF) and one originated from the EF were also acquired commercially. Marshallberg Farm (MF), Smyrna, NC, USA, was the main source of the Russian sturgeon DNA samples that originated either from individual caviar grains from different females or from skin swab samples (derived from both males and females of different ages (1–7 years old)).

### 4.2. Sturgeon Total DNA Isolation

DNA from the individual caviar grains was isolated using an Invitrogen PureLink Genomic DNA Mini Kit (Invitrogen, Waltham, MA, USA) according to the manufacturer’s instructions. Commercially acquired caviar samples were stored in a refrigerator before DNA isolation for no more than 1 week after arrival. DNA was isolated from at least 12 individual caviar grains representing each different fish farm. An in-house, cost-effective skin swabbing method for DNA sample collection from small laboratory fish [19] was successfully adapted for DNA isolation from the sturgeon fish grown at the MF.

### 4.3. Cell Lines and Expression Plasmids

African green monkey kidney (Vero) cells were obtained from the American Type Culture Collection (Manassas, VA, USA) and were maintained in Dulbecco’s modified Eagle’s medium (Gibco BRL, Grand Island, NY, USA) supplemented with 10% fetal calf serum and antibiotics/antimicotics.

Plasmids pcDNA 3.1/V5-His-TOPO and pcDNA 3.1/V5-His-TOPO/LacZ were part of the pcDNA 3.1/V5-His-TOPO Expression Kit (ThermoFisher Scientific, Waltham, MA, USA). Plasmid pBluescript SK (+) containing a large and convenient for subcloning purposes linker region was obtained from Addgene (Watertown, MA, USA). Plasmid pBV-Luc containing the luciferase reporter gene was also obtained from Addgene. This vector is considered to be one of the best for the luciferase reporter assay and was created by a collective effort from the Molecular Genetics Laboratory of the Johns Hopkins Oncology Center in 1998 [20]. The vector features a transcription blocker and convenient multiple cloning sites linker sequences located upstream of the reporter gene. As a result, the basal luciferase activity was supposed to be extremely low. The EGFP expression cassette under the regulation of the HCMV-IE promoter was subcloned into pBluescript SK (+) and used as control (construct #18) for the transfection optimization.

### 4.4. Primer’s Design and PCR Conditions

All PCR reactions were performed using the FailSafe polymerase system (Biosearch Technologies, Hoddesdon, UK) and either buffer B or buffer E ready-to-go pre-mixes. DNA-based sex determination was performed using a recently published AllWSex2 primers set [5] combined with our modification to include *A. gueldenstaedtii*-specific tetrasomic microsatellite loci Ag49 primers [21] to validate the presence of the DNA template and to avoid the situation when no PCR products samples would be automatically assigned as derived from the male fish. All PCR setups included 3 min at a 96 °C step for the initial DNA denaturation, which was followed by 30–35 cycles with 15 s at 94 °C for template denaturation, 30 s at 55 °C for primers annealing, and 45 s at 72 °C for the product extension. A final extension at 72 °C for 7 min was included in all PCR conditions. The annealing temperature was adjusted to 50 °C for the random PCR amplification using short oligonucleotides. The polymerase extension time was also varied based on the size of the expected PCR product (1 min for each 1 kb of the amplified sequence). All primers nucleotide (nt) sequences are shown in Appendix A.

### 4.5. Plasmid DNA Purification, Cloning, and Conventional Sequencing

Multiple recombinant plasmid DNAs were isolated using different scales (large scale for pBluescript SK (+) vector and midi scale for all other plasmids) Qiagen Plasmid Purification Kits (Qiagen, Germantown, MD, USA) based on the expected DNA quantities needed. Some DNA fragments were isolated by agarose gel electrophoresis and purified from the gel using a Zymoclean Gel DNA Recovery Kit (Zymo Research, Irvine, CA, USA). Conventional Senger DNA sequencing was performed by the LSU SVM Gene Lab Core Sequencing Facility. A minimum of 12 DNA samples representing each different fish farm’s caviar were used for PCR and sequencing. Both strands for the small PCR products were sequenced directly using amplification primers. In some instances, DNA was cloned first using the pcDNA 3.1/V5-His-TOPO cloning vector, and plasmids were sequenced using a vector-based CMV forward primer. All enzymes for DNA restriction/modification were obtained from New England Biolabs (Ipswich, MA, USA).

### 4.6. Reconstruction of the ~5.6 kb DNA Fragment Corresponding to A. baerii Genomic Region Preceding IGLV Gene (“Old” A. baerii Nts 3202–8806, GenBank Accession Number AJ245365)

Approximately 5.6 kb DNA fragment spanning the area preceding the *A. baerii* IGLV gene for the immunoglobulin light chain variable region was reconstructed using DNA isolated from the individual EF-derived Siberian sturgeon caviar grain and overlap-extension PCR method as we described earlier [22]. The naturally occurring unique restriction site Bam HI (nt 3558) was part of the forward primer 1F, while a unique restriction site Hind III was added after nt 8806 as part of the assembly reverse primer 2R (Appendix A). Primers 1 and 3 combinations generated a 1725 bp PCR product; primers 4 and 5 produced a 1785 bp PCR product; and primers 6 and 2 generated a 1831 bp PCR product. After purification, DNA fragments were mixed using an equimolar ratio, and a 5225 bp PCR product was generated using outside primers 1 and 2. The final Bam HI-Hind III DNA fragment (nt positions 3558–8806) was subcloned into the pBluescript SK+ vector for sequencing confirmation. The remaining segments (nt positions 3202–3558 or ABter domain) were ordered from IDT (Coralville, IA, USA) as different versions (“old” *A. baerii* sequence, 168 bp variant, 139 bp variant or a complete 168 bp deletion) of G-block DNA fragments that also contained restriction sites EcoR V (nt 3202) and Bam HI (nt 3558) for cloning purposes. Within each G-block, the genomic region that corresponded to the “old” *A. baerii* nts 3361–3558 or ∆ABter was the same, while the segment covering nts 3202–3361 was varied (nt position 3361 corresponds to the last base of the 168 bp long DNA fragment). The DNA fragment covering nts 3558–8806 was derived from the selected clone 9 (out of a total of 12 clones analyzed) during an assembly, cloning and sequencing. Different versions of the EcoR V-Hind III DNA fragment were finally subcloned into plasmid pBV-Luc [20], confirmed by digestion and sequencing, and used for transfection into Vero cells and luciferase expression detection.

Six different plasmids were initially constructed, all of which (except for the pBV-Luc control plasmid #1) contained the same sequence derived from assembly clone 9 (nucleotides 3558–8806), while the respective variations were introduced within the nucleotides covering the 3202–3558 region. Plasmid #2 included the original “old” *A. baerii* sequence spanning nucleotides 3202–3558. Plasmid #3 contained the 168 bp version, which preserved the overall sequence size but incorporated all corresponding Russian sturgeon nucleotide substitutions within the region spanning nucleotides 3202–3361. Plasmid #4 represented the 139 bp version, which featured a 29 bp deletion along with all the Russian sturgeon-specific mutations. Plasmid #5 was engineered with a complete deletion of the 168 bp DNA fragment (nucleotides 3202–3361), and plasmid #6 had the entire region spanning nucleotides 3202–3558 deleted. Two more plasmids were also engineered, each containing a 569 bp DNA fragment covering nucleotides 8337–8806, which presumably harbored the native IGLV gene promoter. Plasmid #7 represented the original “old” *A. baerii* sequence published 25 years ago, while plasmid #8 contained the same-sized sequence derived from the assembly clone 9.

To clarify the possible role of the repetitive portion of this *A. baerii*-specific genomic region for the luciferase expression inhibition, we have constructed a positive control luciferase expression plasmid that worked efficiently after transfection in the Vero cells (pBV-Luc-CMV or construct #9). This plasmid contained the HCMV-IE promoter sequence (nucleotides 236–852 from the pcDNA 3.1/V5-His-TOPO vector), which was inserted into the pBV-Luc plasmid after the existing transcription blocker and before the luciferase gene. Plasmid #10 carried the entire EcoR V-Bam HI fragment (nucleotides 3202–3558 or ABter) from the original “old *A. baerii*” sequence, which was cloned into pBV-Luc in the positive (+) orientation after the transcription blocker but before the luciferase gene. Plasmid #11 featured the same sequence in the positive (+) orientation, which was cloned into the pBV-Luc-CMV plasmid after the transcription blocker but preceding the HCMV-IE promoter sequence. Plasmid #12 contained the same DNA fragment (ABter) but was cloned in the opposite (−) orientation, and it was positioned after the HCMV-IE promoter and before the luciferase gene. Construct #13 was like #12 except that the ABter region was in the positive (+) orientation. Plasmids #14 through #17 were constructed using a similar approach; however, only the genomic region spanning nucleotides 3361–3558 (∆ABter) was used for cloning. This fragment did not include the variable portion of the repeat found in different sturgeon species. Finally, plasmid #18 was pBluescript-based that contained an EGFP expression cassette (HCMV-IE-EGFP-polyA) and was used as a control to optimize the transfection efficiency.

### 4.7. DNA Transfection, Cell Lysis, β-Galactosidase and Luciferase Detection

Twenty-four hours before transfection, Vero cells (10^5^ cells) were seeded in 6-well plates to have 80–90% confluency. Lipofectamine 3000 (ThermoFisher Scientific, Waltham, MA, USA) was used for transfection as described by the manufacturer’s instructions to study the level of luciferase expression. Different constructs were co-transfected with pcDNA 3.1/V5-His-TOPO/LacZ control (DNA total is 5 µg/well and each containing 2 µg of LacZ control and 3 µg of the DNA representing the respective test construct). This ratio was empirically determined based on the sensitivity of the β-galactosidase detection assay. The LacZ control plasmid was used to normalize the possible differences between the samples during luciferase expression level detection. Every plasmid construct used for transfection was represented by two different individual biological replica clones, and the results were adjusted considering the level of the β-galactosidase expression together with the total protein concentration within the cell lysates.

Cells were collected 48 h post-transfection, washed 2× with 1 mL of ice-cold PBS (w/o Ca^2+^ and Mg^2+^) at 1000× *g* for 5 min and resuspended in 120 µL of PBS. Then, 30 µL of 5× reporter Lysis Buffer (Promega, Madison, WI, USA, Cat. # E3971) was added to the cell suspension, and 50 µL of lysates was used for luciferase detection using a ONE-Glo™ EX Luciferase Assay System (Promega, Madison, WI, USA, Cat. # E 8110) as described by the manufacturer’s instruction (12.5 μL for each of the 4 replicas). The remaining 100 µL of lysates was subjected to 2 cycles of freeze/thaw at −80 °C for the 30 min freezing/10 min thawing in ice-cold water, which was clarified by centrifugation (at 16,100× *g* for 10 min) before assays, and the same lysates were used for both the β-Gal Assay Kit (Promega, Madison, WI, USA, Cat. # K1455-01) and Pierce BCA Protein Assay Kit (Pierce, Appleton, WI, USA, Cat. # A55865, 0.125–2.0 mg/mL standard curve range; 1.0–1.6 mg/mL samples range) to normalize all the luciferase expression results generated after using a Cytation 5 Cell Imaging Multi-Mode Reader (BioTec Instruments, Inc., Minneapolis, MN, USA). We did not observe any significant variation in β-galactosidase activity between samples (around 25% maximum variations), but the corresponding numbers were still used for normalization.

### 4.8. Statistical Analysis

Quadruplicate luminescence values were used in one-way ANOVA analysis with Tukey’s test for multiple hypothesis testing. GraphPad Prism 10 software was utilized to perform statistical analysis and generate bar graphs.

## 5. Conclusions

The present study aimed to investigate the role of the repetitive sequences within the Russian sturgeon genome. It was shown that both unchanged over the years and mutated copies of the same genomic region specific for the multiple *Acipenser* species and located within the sequence preceding the *A. baerii* IGLV gene cluster exist simultaneously within the same organism regardless of the fish’s geographic origin, sex, age, or source of DNA isolation (individual caviar grains or skin swab samples). The process of selection toward specific mutations appears not random and is ongoing based on the sequence variations within DNA derived from the different caviar grains but originated from the same fish. We report that specific for different *Acipenser* species repetitive region within the sequence preceding the *A. baerii* IGLV gene can serve as a transcription termination element. Increased variability within this genetic locus may be predictive of the ability of certain fish species to combat infections, therefore increasing their survivability.

## Figures and Tables

**Figure 1 ijms-25-12685-f001:**
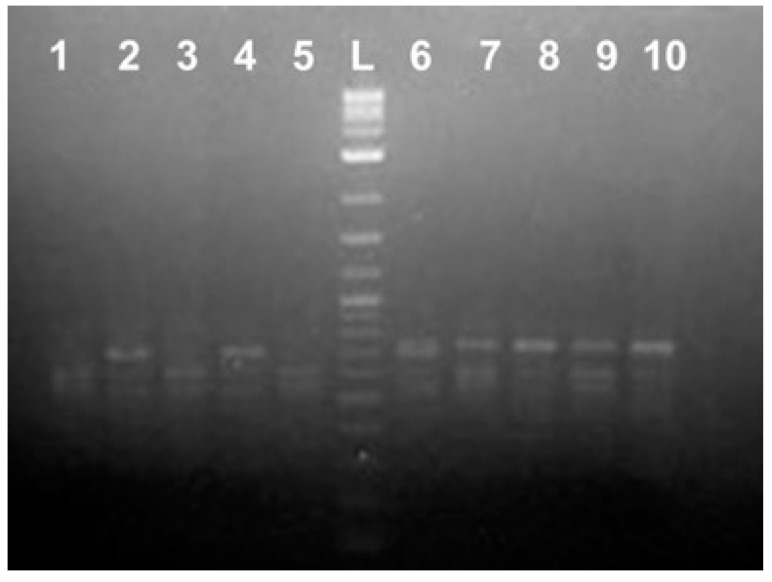
Random PCR amplification using DNA from the adult Russian sturgeon (*Acipenser gueldenstaedtii*). Lanes 1–5: adult females (F1 through F5) DNA; 6–10: adult males (M1 through M5) DNA; L: 2-log DNA ladder (NEB).

**Figure 2 ijms-25-12685-f002:**
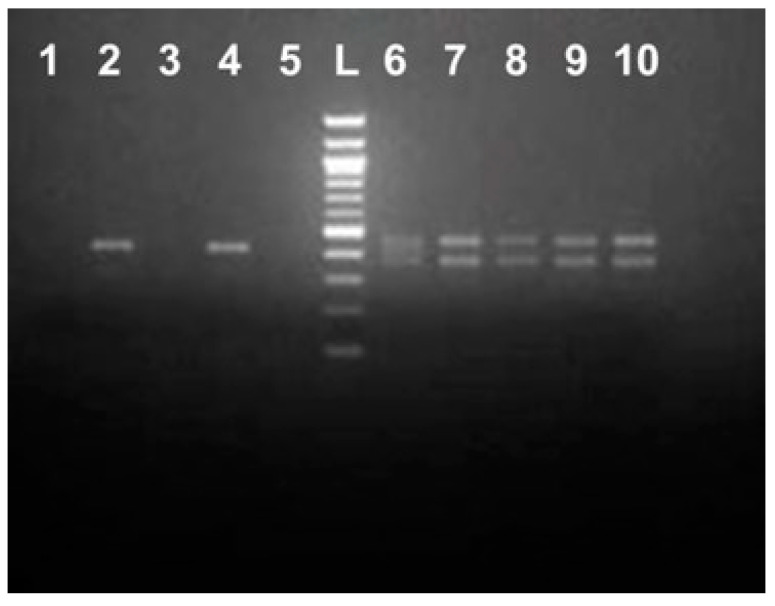
Specific PCR amplification using primers based on the VAC-1M sequence and DNA from the adult Russian sturgeon (*Acipenser gueldenstaedtii*). Lanes 1–5: adult females (F1 through F5) DNA; 6–10: adult males (M1 through M5) DNA; L: 2-log DNA ladder (NEB).

**Figure 3 ijms-25-12685-f003:**
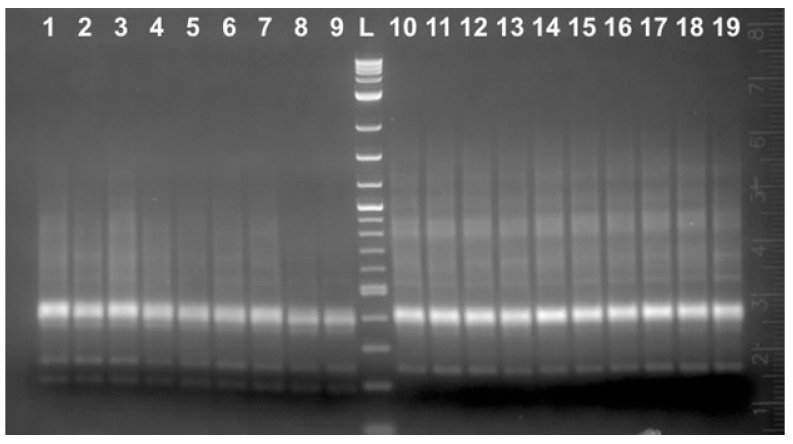
PCR amplification using M5-1F/M5-1Rn primers. DNA was isolated from the individual caviar grains originated from the Russian sturgeon (*Acipenser gueldenstaedtii*) grown in Israel (IF, lanes 1–9) and North Carolina, USA (MF, lanes 10–18); L: 2-log DNA ladder (NEB).

**Figure 4 ijms-25-12685-f004:**
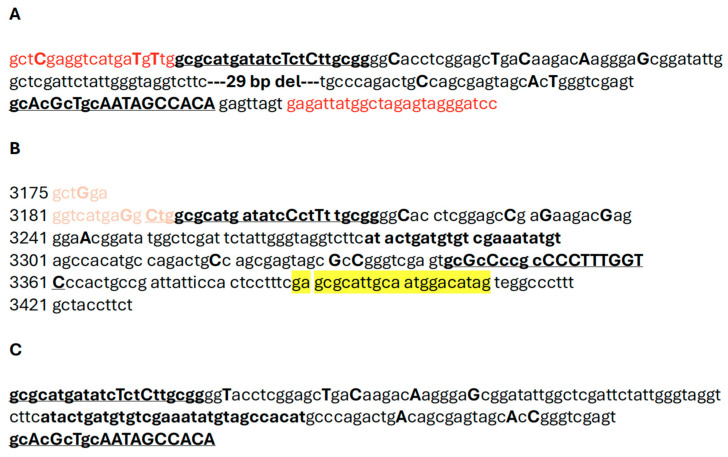
(**A**) Consensus sequence of the 192 bp long DNA (B-fragment). Sequence was generated from the Russian sturgeon (*Acipenser gueldenstaedtii*) grown in Israel. Amplification primers M5-1F and M5-1Rn were found to be a part of the PCR product and are shown in red. All nts that are different between A. gueldenstaedtii and corresponding A. baerii genomic sequences are shown in bold as capital letters. New specific internal primers B-IF-for/B-IF-rev (shown in bold and underlined) were designed based on the 192 bp sequence with the expected size of the PCR product to be 139 bp. (**B**) Siberian sturgeon (*Acipenser baerii*) genomic region (“old” bases 3175–3430; Gene Bank accession number AJ245365). The respective 29 bp nt sequences found to be deleted for *A. gueldenstaedtii* IF-derived PCR products are shown in bold and italics. The predicted size of the PCR product using B-IF-for/B-IF-rev primers is 168 bp. The reverse primer A ba-3411R: ACTATGTCCATTGCAATGCGCTC is shown in yellow. (**C**) Consensus sequence of the 168 bp long DNA fragment. The sequence was generated using amplification primers B-IF-for (corresponds to the respective “old” *A. baerii* nts 3194–3215) and B-IF-rev (corresponds to the respective “old” *A. baerii* nts 3361–3343) and caviar DNA from the *A. gueldenstaedtii* grown at the MF, North Carolina, USA.

**Figure 5 ijms-25-12685-f005:**
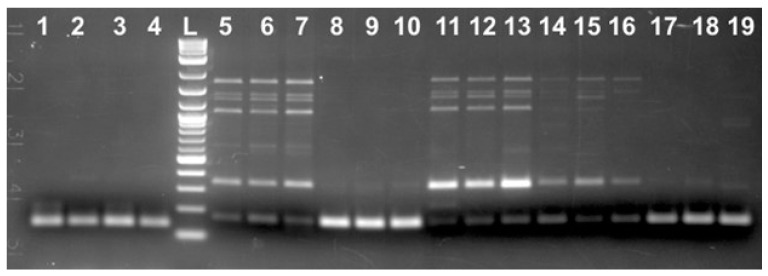
PCR amplification using B-IF-for/B-IF-rev primers and Russian sturgeon (*Acipenser gueldenstaedtii*) caviar DNA from the IF and 5 different female Russian sturgeons from the MF. Lanes 1–4 represent individual caviar grains DNA originated from the Russian sturgeon grown in Israel, and lanes 5–19 represent different Russian sturgeon females (3 lanes each): I, II, III, IV, and V, respectively, grown at the MF, North Carolina, USA. L: 2-log DNA ladder (NEB).

**Figure 6 ijms-25-12685-f006:**
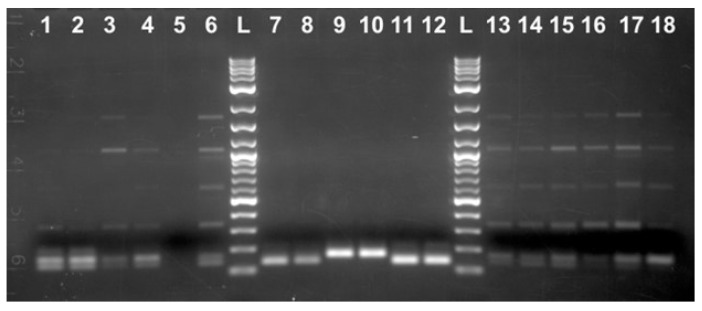
PCR amplification using B-IF-for/B-IF-rev primers and DNA from both Russian (*Acipenser gueldenstaedtii*) and Siberian sturgeon (*Acipenser baerii*). Lanes 1–6 and 13–18 represent individual caviar grains DNA from the Siberian sturgeon (*Acipenser baerii*) grown at the EF, Florida, USA; lanes 7–8 represent caviar DNA from the Russian sturgeon grown in Israel, IF; lanes 9–10 represent caviar DNA from the Russian sturgeon grown at the MF, North Carolina, USA; lanes 11–12 represent caviar DNA from the Russian sturgeon grown at the EF, FL, USA. L: 2-log DNA ladder (NEB).

**Figure 7 ijms-25-12685-f007:**
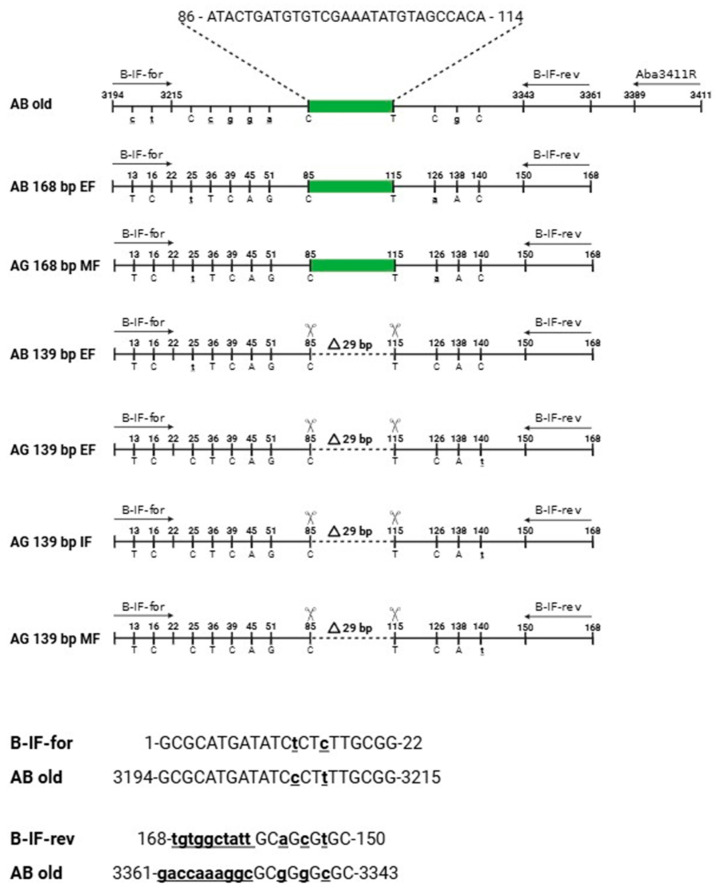
Variable domain of the VAC-2M repetitive region (respective “old” Siberian sturgeon (*Acipenser baerii*) nts 3194–3361): original 168 bp “old” Siberian sturgeon (*Acipenser baerii*) sequence vs. recent 168 bp and 139 bp DNA fragments from the Siberian and Russian sturgeon (*Acipenser gueldenstaedtii*). AB old represents the respective sequence originated from the *A. baerii* as reported in AJ245365; AB-represents recent sequences originated from the *A. baerii*; AG-represents the respective recent sequences originated from the *A. gueldenstaedtii*. Nucleotides that are different between the corresponding sequences are shown in bold as small and underlined letters.

**Figure 8 ijms-25-12685-f008:**
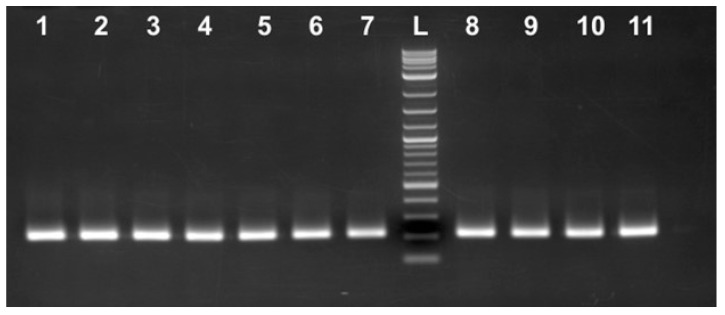
PCR amplification using B-IF-for/A ba 3411R primers. Lanes 1–3 represent individual caviar grains DNA from the Starry sturgeon (*Acipenser stellatus*) grown at the EF; lanes 4–5 represent caviar DNA from the Siberian sturgeon (*Acipenser baerii*) grown at the EF; lanes 6–7 represent caviar DNA from name the Russian sturgeon (*Acipenser gueldenstaedtii*) grown at the EF; lanes 8–9 represent caviar DNA from the *A. gueldenstaedtii* grown at the MF; lanes 10–11 represent caviar DNA from the *A. gueldenstaedtii* grown in Israel. L: 2-log DNA ladder (NEB).

**Figure 9 ijms-25-12685-f009:**
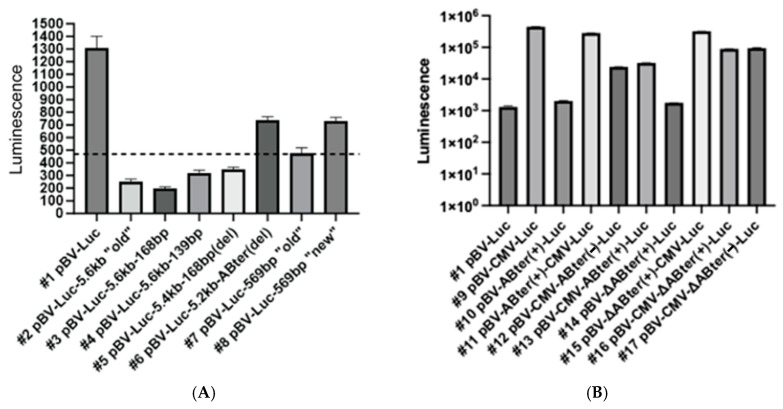
(**A**) Luciferase expression levels in Vero cells using constructs containing different versions of ~5.6 kb sequence preceding IGLV gene cloned into pBV-Luc plasmid. Plasmids with “old” abbreviation represent sequences derived from the GenBank AJ245365 reference Siberian sturgeon (*Acipenser baerii*) reported 25 years ago, while “new” denotes the sequence variant detected recently. (**B**) Luciferase expression levels in Vero cells using constructs containing ABter (“old” Siberian sturgeon (*Acipenser baerii*) nts 3202–3558) and ∆ABter (“old” *A. baerii* nts 3361–3558) cloned under control of the HCMV-IE promoter into pBV-Luc plasmid. (+) denotes positive DNA strand orientation, while (−) is an indication that the same sequence was cloned in the opposite orientation.

## Data Availability

The raw data supporting the conclusions of this article will be made available by the authors on request.

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
