# Peer review of "A Repetitive Acipenser gueldenstaedtii Genomic Region Aligning with the Acipenser baerii IGLV Gene Cluster Suggests a Role as a Transcription Termination Element Across Several Sturgeon Species"

_ijms, 2024, doi:10.3390/ijms252312685_

Round 1
Reviewer 1 Report
Comments and Suggestions for Authors
In my opinion, MS raises really important aspects for sturgeon aquaculture. This study focuses on the presence of repetitive sequences in the sturgeon genome, which can enhance immune responses against infectious diseases. In the DNA of Russian sturgeon, the authors identified a repetitive sequence VAC-2M, which shows a match with the IGLV gene region of Siberian sturgeon. MS showed that the diversity of this sequence in different species suggests its conserved nature, which indicates a potential defense mechanism. Studies of the function of this sequence in expression experiments showed that it can act as a transcription terminator, influencing the production of antibodies in sturgeon, which in my opinion is very important and interesting. I believe that the results obtained by the authors in this study can contribute to a better understanding of the mechanisms of immunity in fish and have the potential for further use in immunological studies. Although the article presents interesting and important content, the weakest link of this study is the Discussion section. This section needs to be reinforced with specific references. My other comments are highlighted in the MS text.

Reviewer 2 Report
Comments and Suggestions for Authors
This study provides valuable insights into the complexity of the sturgeon genome and the potential significance of repetitive elements. However, including more details would strengthen the clarity and reproducibility of the study.
Abstract: clearly outlines the study's objective, which is to investigate the role of repetitive sequences in sturgeon immune response.
Are there specific environmental factors or immune challenges that may influence the expression or activity of this sequence?
Introduction:
Consider adding a concise thesis statement at the end of the introduction to clearly outline the primary objective of the study.
Results:
The observed differences in PCR product sizes between sturgeon populations from different geographic locations (Israel vs. Marshallberg Farm) indicate potential regional variation in these repetitive elements. This variation may be driven by factors such as genetic drift, natural selection, or hybridization. A more comprehensive analysis of sturgeon populations from diverse geographic regions is necessary to elucidate the extent of this variation.
Are these mutations spontaneous or induced by environmental factors, such as water quality or temperature?
How does the genomic context of these repetitive regions influence mutation rates and selection pressures?
The functional implications of these mutations remain unclear. Do they affect gene expression, protein function, or other cellular processes?
Could these mutations contribute to phenotypic variations, such as growth rate, disease resistance, or reproductive success?
Are there specific methods to optimize DNA extraction from caviar samples?
Discussion:
Are there other functional implications of the VAC-2M sequence, such as involvement in DNA repair, recombination, or epigenetic regulation?
The authors mention using β-galactosidase expression for normalization. Was there any significant variation observed in β-galactosidase activity between samples?
Lines 520-526: Explain how the results of the study can be used in future applications of aquaculture.
Material and methods:
Lines 529-535: The text doesn't specify the number of samples used from each source/species. There's no information about how the samples were stored, transported, or handled before DNA analysis. There's no description of the sampling protocol used for collecting the skin swab samples or extracting caviar samples. Standard protocols should be detailed for reproducibility. For the skin swab samples, while it mentions they came from "different ages," it doesn't specify the age ranges. This lack of precision could affect result interpretation.
Lines 575-576: specify which exact scales/kit versions were used.
Line 579: PCR Conditions Missing. Clarify Sequencing reaction conditions, Quality score cutoffs, How many replicates were performed.
Lines 644-646: Good detail on cell preparation and transfection reagent, but missing specific cell counts seeded per well.
Lines 647-650: The ratio of DNA amounts (3:2) is specified, but should explain why this ratio was chosen for LacZ control vs test construct.
Line 651: States "Every plasmid construct" but doesn't specify the total number of constructs used in the study.
Lines 652-654: Mentions "two different individual clones" but lacks explanation of clone selection criteria or verification method.
Lines 655-657: Cell collection protocol is well detailed with specific volumes, but missing centrifugation speed and time for the washing steps.
Lines 657-659: missing the volume of cell suspension used per assay.
Lines 659-661: The freeze/thaw cycle description lacks critical parameters: Duration of freezing, Duration of thawing, Temperature during freezing
Lines 661-665: No standard curve details provided and protein concentration acceptance range.
Round 2
Reviewer 1 Report
Comments and Suggestions for Authors
Thank you for taking into account my suggestions regarding the manuscript.
Please correct the Latin names of species (bye example: Acipenser gueldenstaedtii) in the entire MS (in the descriptions of tables, figures) - they should be written in italics.
